# A Simple and Effective Method to Adjust the Structure and Performance of DLC Films on Polymethyl Methacrylate(PMMA) Substrate

**Yinzhong Bu [1], Kaihuan Yu [2], Bin Zhang [2,3], Bin Kuang [1,\*] and Li Qiang [2,3,\*]**

[1] Department of Stomatology, The First People's Hospital of Lanzhou City, Lanzhou 730050, China
[2] Key Laboratory of Science and Technology on Wear and Protection of Materials, Lanzhou Institute of Chemical Physics, Chinese Academy of Sciences, Lanzhou 730000, China
[3] Center of Materials Science and Optoelectronics Engineering, University of Chinese Academy of Sciences, Beijing 100049, China
[\*] Correspondence: kuangbin1223@163.com (B.K.); qiangli1413@licp.cas.cn (L.Q.); Tel.: +86-0931-2337286 (B.K.); +86-0931-4968005 (L.Q.)

**Abstract:** DLC (diamond-like carbon) films were prepared on Ti/PMMA(polymethyl methacrylate) under the different bias; the principal purpose of this study is to explore structural differences of films on Ti/PMMA with and without conductive material, and to further clarify the role of the conductive Ti interlayer in influencing the deposition mechanism. The results indicate that the films deposited on Ti/PMMA with conductive material exhibit uniform DLC structure and mechanical hardness when the bias voltage is ≥550 V. Furthermore, the deposited DLC does not change the wettability of PMMA, while the addition of the Ti interlayer virtually increases the risk of peeling off of the film. The results of the tribological study demonstrate the films on Ti/PMMA with conductive material have better tribological properties than those without conductive adhesive. This research work can provide basic theoretical guidance for depositing uniform DLC films on PMMA and even on all non-conductive substrates.

**Keywords:** DLC films; PMMA denture base; Ti interlayer; microstructure; tribological behaviors




## 1. Introduction

Since Walter Bauer first applied polymethyl methacrylate (PMMA) to the fabrication of dentures in 1937 [1], the thermosetting PMMA has been used widely as a denture base material in clinical practice due to its low price, good biocompatibility, easy processing, and repair [2]. However, this material cannot fully meet the requirements of denture repair due to its poor mechanical hardness [3]. Poor mechanical hardness makes PMMA easy to scratch when making contact with hard objects, and thus its surface roughness will increase after long-term use. While high roughness will lead to the easy adhesion of pathogenic bacteria on its surface, which can cause dental caries, periodontitis, denture stomatitis, and other oral diseases. Therefore, scholars have carried out a lot of research to enhance the mechanical properties of PMMA. For example, the addition of various fibers in PMMA can enhance its mechanical properties, but these fibers have their own shortcomings, such as inconvenient operation, high brittleness, and other problems, which restrict their development and application [4,5]. Therefore, it is necessary to effectively improve the mechanical properties of PMMA to increase the service life of the denture.

The surface modification technology can maintain a series of excellent qualities of PMMA as the matrix material, and greatly improves its comprehensive performance. Therefore, surface modification has become one of the fastest growing fields of biomedical materials [6]. Diamond-like carbon (DLC) film is a kind of amorphous carbon film material, in which carbon atoms are hybridized by sp3 and sp2 [7], and thus it has both the hardness

of diamond and the tribological properties of graphite. As a result, it has received extensive attention in the fields of machinery, optoelectronics, and magnetic medium protection because of its high hardness, excellent wear, and corrosion resistance [8]. At present, many experiments in vivo and in vitro have proved that DLC films also have excellent biocompatibility and blood compatibility. Therefore, more and more people pay more attention to the application of DLC films in the biomedical field, including artificial heart valves, medical wire guides, plastic surgery instruments, etc. [9–11].

As mentioned above, DLC film was attempted to be prepared on the surface of PMMA, hoping to better combine the characteristics of polymethyl methacrylate and DLC, and seeking an effective way to improve the mechanical properties of a denture base resin surface. However, the deposition results were not satisfactory in the first phase of the experiment. That is, it is difficult for DLC films to be directly deposited on the surface of PMMA. Therefore, in this work, a simple and effective method was provided to prepare uniform DLC on the surface of PMMA; the principal purpose of this study is to explore structural differences of films on Ti/PMMA with and without conductive material, and to further clarify the role of the conductive Ti interlayer in influencing the deposition mechanism of DLC film on PMMA. This research work can provide basic theoretical guidance for depositing uniform DLC films on PMMA and even all non-conductive material by the plasma-enhanced chemical vapor deposition (PECVD) method.

## 2. Experimental Details

### 2.1. Film Deposition

Pink PMMA was firstly cut into $20 \times 20$ mm$^2$ and adopted as substrates. All substrates were ultrasonically pre-cleaned with soapy water and hot deionized water in order to remove the contaminations.

The first stage experiment: DLC film was attempted to be deposited directly on the PMMA substrate by PECVD. Prior to the deposition, the chamber was evacuated up to $1.0 \times 10^{-3}$ Pa, and all PMMA substrates were pretreated by an argon discharge for 5 min. During the deposition of DLC film, the flow rate of CH$_4$ maintained a constant value of 20 sccm, while the negative substrate bias was 950 V, powered by a DC(direct-current) superposition pulse bias power supply (Chengdu Pulsetec Electric Co., Ltd., Chengdu, China). The working pressure was about 17–21 Pa, and the deposition time was kept at 60 min for all the samples. Then, the Raman was conducted to measure the microstructure of the films, and it was found that the Raman peak shape of the sample was basically similar to that of PMMA. Therefore, it was determined that it was very difficult to directly deposit DLC film on PMMA using PECVD in this work.

The second stage experiment: The experimental scheme was adjusted. The Ti interlayer was deposited on PMMA by magnetron sputtering. During the deposition, the flow rate of Ar was 45 sccm, the sputtering power of Ti target was about 1.0 kW (~500 V, 2.0 A), and the deposition time was 15 min. Then, the sample was taken out of the magnetron sputtering vacuum chamber quickly and put into the PECVD vacuum chamber immediately (Figure 1). Additionally, the conductive materials (silver conductive adhesives) were bonded on the surface of some samples to ensure that the Ti interlayer was connected with the steel sample holder. During the deposition of DLC top layers, the flow rate of CH$_4$ was maintained at a constant value of 20 sccm, while the substrate negative bias changed from 350~950 V, powered by a DC superposition pulse bias power supply (Chengdu Pulsetec Electric Co., Ltd., Chengdu, China). The working pressure was about 17–21 Pa, and the deposition time was kept at 60 min for all the samples. After deposition, the sample was cooled down to room temperature inside the chamber in an argon atmosphere. In the further text, sometimes the terms "DLC350V-D, DLC550V-D, DLC750V-D, DLC950V-D" were used to denote the films deposited with conductive materials, while "DLC350V-BD, DLC550V-BD, DLC750V-BD, DLC950V-BD" were used to denote the films deposited without conductive materials at substrate bias of $-350$ V, $-550$ V, $-750$ V, and $-950$ V on

Ti/PMMA, respectively. Furthermore, a minus sign before specific bias was not added in order to avoid misunderstanding in the next text.

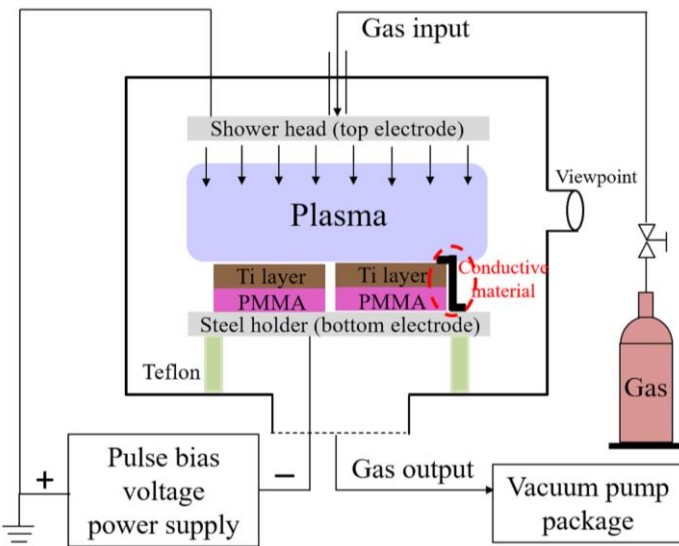

**Figure 1.** The schematic diagram of PECVD system for DLC films deposited on Ti/PMMA with and without conductive materials.

### 2.2. Sample Characterization

The chemical bonding of the films was examined by using a LabRAM HR Evolution Raman Spectroscope (HORIBA Jobin Yvon S.A.S., Palaiseau, France) at the excitation wavelength of 532 nm and with a spot size of 2 μm. The integration time was 20 s, the accumulations was 3. The mechanical properties were measured by a nanoindenter (Nano indenter II, MTS Co., Ltd., USA) with a maximum indentation depth of 50 nm in order to avoid the influence of the substrate. A DSA100 CA measurement instrument (KRUSS, Hamburg, Germany) was used to measure the static CA, where a droplet of 1 μL (diiodomethane and water) was adopted by the sessile drop method. The static CA value at steady state after 15 s was collected for each of the measurements.

The adhesive strength of films on PMMA was measured by the Scratch testing system. The samples were scratched using a cone-shaped diamond indenter (HRC-3, R = 0.2 ± 0.01 mm, a = 120 ± 21°), and the test load was monotonously increased from 0 to 15 N during the measurement. The scratch speed and its length were constant at 15 N/min and 5 mm, respectively.

The tribological performance was conducted at room temperature (23 °C) on self-developed friction and wear tester with a rotating ball on-disc configuration selecting commercial ø5 mm GCr15 steel balls as counterparts. The parameters of tribotests were as follows: the frictional load was 1 N, and the rotational speed was fixed to 300 rpm with a radius of 3 mm. The testing time was 30 min and a constant humidity of 27 ± 1% was maintained with a humidity regulator. All tribotests were carried out at least three times to ensure the repeatability. After testing, the surface morphology of wear tracks of samples was characterized with an OLYMPUS optical microscope.

### 3. Results and Discussion

#### 3.1. Raman Analysis

Raman spectroscopy is usually used to measure the detailed structural characterization of the DLC film as a nondestructive tool. Figure 2 displays the Raman spectra of PMMA, Ti/PMMA, and DLC prepared on Ti/PMMA at a different substrate bias without conductive materials. It can be seen that the Raman curve is wavy for the original PMMA, which is the typical Raman spectrum of the polymer [12]. For the Ti layer, no Raman peak can be observed because its peak intensity is too weak to be observed. For the films on

Ti/PMMA, the film exhibits the same Raman characteristics as the Ti layer when the bias voltage is 350 V, indicating the failure of film deposition. With the increase of bias voltage to 550 V and 750 V, the films exhibit a similar Raman curve as the PMMA substrate no matter what the test position is in the center or at the edge of the sample, implying a polymer-like structure [13]. However, when the bias voltage finally increases to 950 V, the Raman peak shape has changed obviously at different positions. When the test position is in the center of the sample, the film is a polymer-like film. As the test position gets closer to the edge, a clear peak can be found at 1500 cm$^{-1}$, and this peak can be fitted as two Gaussian peaks located at around 1550 cm$^{-1}$ (G peak) and 1350 cm$^{-1}$ (D peak), which indicates a typical characteristic of DLC films [14]. Addtionally, the closer to the edge, the stronger peak intensity of the film.

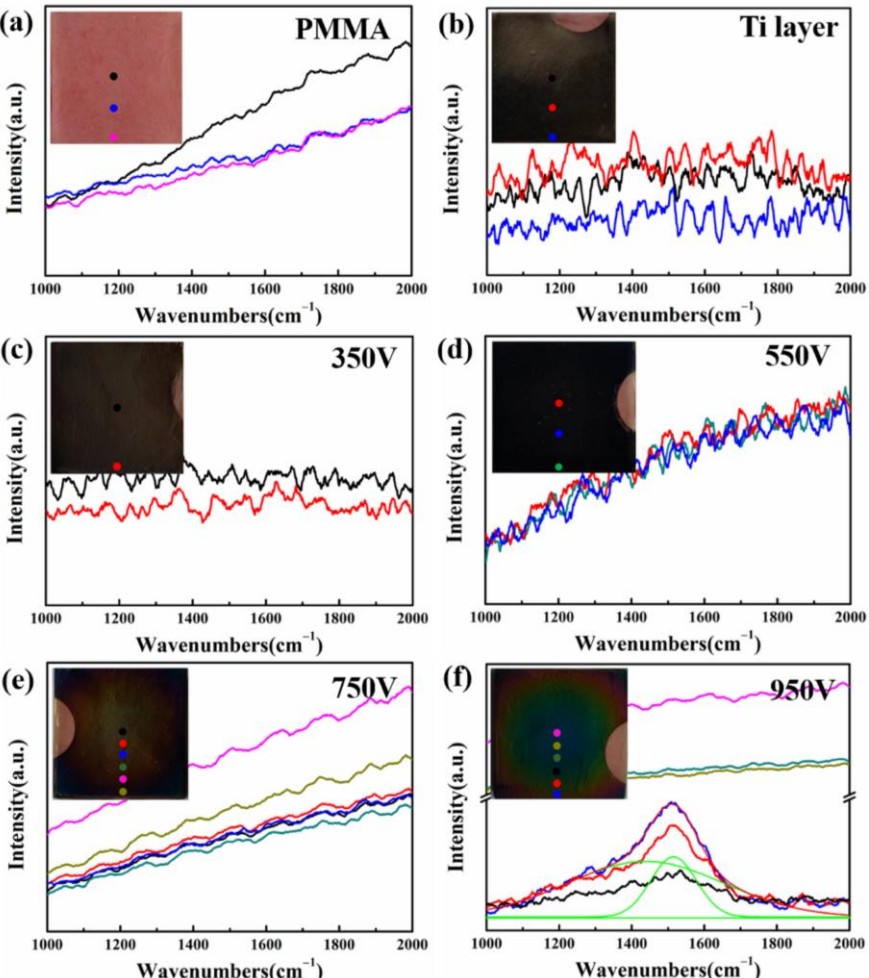

**Figure 2.** Raman spectra measured at different positions of (**a**) PMMA, (**b**) Ti/PMMA and the films prepared on Ti/PMMA at different substrate bias of (**c**) 350 V, (**d**) 550 V, (**e**) 750 V, and (**f**) 950 V without conductive material. The color of the Raman curve represents the results measured at the corresponding color position in the illustration.

Figure 3 displays the Raman spectra of films prepared on Ti/PMMA at a different substrate bias with conductive materials. When the substrate bias is 350 V, no Raman peak could be found, which implies the failure of films (polymer-like films or DLC films) deposition. However, as the bias voltage increases from 350 V to 550 V, the DLC550V-D film exhibits the typical DLC structural characteristic no matter whether the test position is in the center or at the edge of the sample. As the bias voltage increases further (≥750 V), the same phenomenon can be clearly observed. These results indicate that when the bias voltage is lower than 350 V, the film deposition always fails regardless of whether there

is conductive material or not. When the bias voltage is greater than 550 V, there is a great difference in the structure of films deposited on Ti/PMMA with or without conductive materials. In addition, when the bias voltage is higher (950 V), the structure of the center and edge of the films on Ti/PMMA without conductive material is also obviously different.

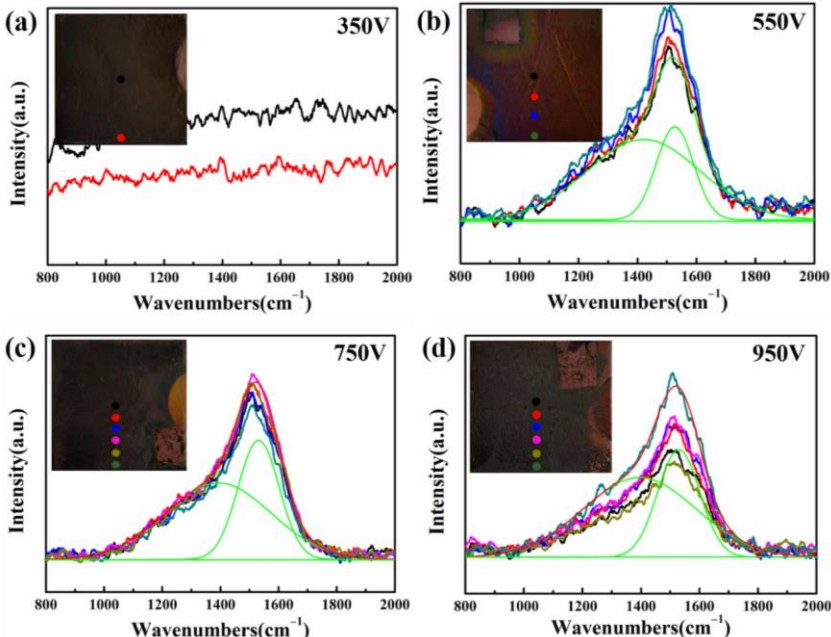

**Figure 3.** Raman spectra of the films prepared on Ti/PMMA at different substrate bias of (**a**) 350V, (**b**) 550V, (**c**) 750V and (**d**) 950V with conductive material. The color of the Raman curve represents the results measured at the corresponding color position in the illustration.

In conclusion, when the bias voltage is ≤350 V, no film (polymer-like or DLC film) can be deposited on PMMA or Ti/PMMA with or without conductive materials. When the bias voltage is ≥550 V, the films deposited without conductive materials are polymer-like films, and even when the bias voltage is high enough (950 V), Raman characteristic peaks of the DLC films are detected only at the edge of the sample. For the films deposited with conductive materials, the films are DLC films no matter whether they are in the center or at the edge of the sample. Why is the film structure so different? This may be closely related to the deposition mechanism of films (Figure 4).

In this work, the capacitively coupled PECVD is used for the deposition of thin films. The capacitor structure is composed of two parallel electrode plates, and an electric field (powered by Pulse bias power supply) is applied between the electrode plates. When the working gas (methane in this work) is introduced into the vacuum chamber, the high electric field between the plates generates glow discharge to decompose and ionize the working gas, forming many electrons, ions, and active groups such as free radicals in the vacuum chamber; these ions collide with each other and flow directionally under the action of an electric field, and enrich on the surface of the substrate to form the film.

When the bias voltage is ≤350 V, the electric field between the two plates is not enough to generate glow discharge to ionize the working gas; this is confirmed by the fact that glow is hardly observed through the viewpoint on the vacuum chamber. Namely, the gas is eliminated without ionization, leading to the failure of film deposition. This is the main reason why it is impossible to deposit films on Ti/PMMA with or without conductive material under a lower bias.

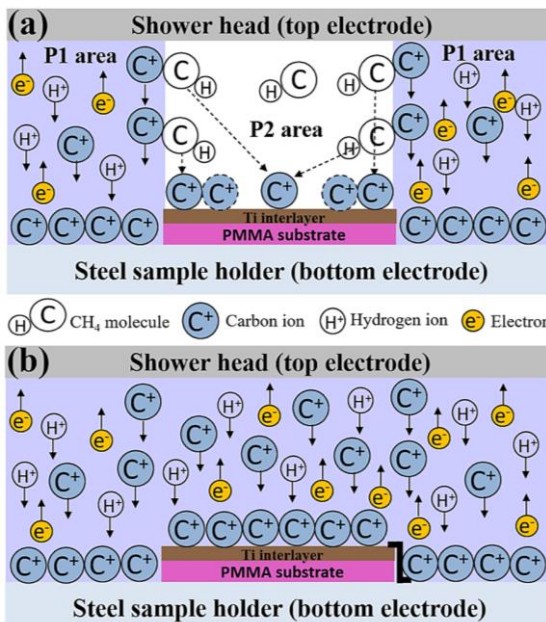

**Figure 4.** Schematic diagram of deposition mechanism of films deposited on Ti/PMMA with (**b**) and without (**a**) conductive material.

For the films deposited on Ti/PMMA without conductive material, as the bias voltage increases to 550 V or 750 V, the electric field between the two plates is high enough to generate the plasma. However, the electric potential field strength between the top electrode and Ti/PMMA substrate is still zero because of good insulation of the PMMA substrate. That is to say, the area between the top plate and PMMA (P2 area as shown in Figure 4a) does not generate plasma and flow directionally. Nevertheless, the electric potential field strength between the top plate and steel sample holder increases with the increase of bias voltage, which means that these high energetic particles in the P1 area may collide with the unionized gas between the top plate and Ti/PMMA and exchange energy, resulting in the change of carbon ionic energy reaching the Ti/PMMA surface. At this moment, the deposition of the film mainly depends on the collision of the ions between the top plate and bottom plate (steel sample holder), and the unionized gas between the top plate and Ti/PMMA to generate ions, while the energy of these ions is too low to form DLC films. As the bias voltage increases to 950 V further, the electric potential field strength between the top plate and steel sample holder increases further with the increase of bias voltage, which means that the ion energy transferred to the substrate surface for film deposition is further increased. Moreover, the energy of film deposition mainly depends on high-energy ion collisions and energy exchange. When the exchange energy is the same, the transverse distance of ions to the center of the substrate is significantly greater than that to the edge of the substrate. As a result, the energy when they reach the center of the substrate surface is higher than when they reach the edge surface of the substrate. Namely, the ion energy reaching the edge of the sample is obviously different from that reaching the center, resulting in the difference of the film structure between the center and edge of the sample.

For the films deposited with conductive material, because the conductive material connects the Ti layer with the steel sample holder (Figure 4b), the electric potential field strength between the top plate and Ti/PMMA is no longer constant, but increases with the increase of bias voltage due to the good conductivity of the Ti interlayer. That is, the ion energy between the top plate and Ti/PMMA is almost the same as that between the top plate and steel sample holder, and there is no difference in the ion energy reaching the center and edge of the sample. As a result, DLC films with uniform structure were successfully prepared on PMMA under a higher bias (≥550 V). Furthermore, it is worth mentioning that

this simple method can be used to effectively prepare carbon films with uniform structure on the surface of all non-conductive materials using the PECVD technology in this work.

### 3.2. Water Contact Angle

The contact angle is an important indicator to reflect the wettability of the material surface. The change of the wettability of the denture base surface may affect its retention force and the adhesion of hydrophobic fungi (such as Candida albicans), triggering bacterial infection and local inflammation [15], which is of great significance to the clinical use of the denture base.

The wettability of the films was assessed using the water contact angle. If the static contact angle is $\theta < 90°$, the material behaves hydrophilic; if the static contact angle is $90° < \theta < 180°$, the material behaves hydrophobic. The results of the contact angle measurement are shown in Figure 5. It can be found that the contact angles of the PMMA substrate are $61 \pm 0.5°$, showing hydrophilicity. When the DLC films are deposited on PMMA with conductive material under a higher bias ($\geq$550 V), the contact angle decreases slightly to $51 \pm 2°$ for the DLC550V-D film, $55.7 \pm 0.1°$ for the DLC750V-D film, and $58.6 \pm 0.7°$ for the DLC950V-D film, respectively. Moreover, there is no obvious difference between the contact angle of the center and the edge of the sample, which may be attributed to the structural uniform of DLC films on Ti/PMMA confirmed by Raman analysis. The above experimental results indicate that the deposited DLC does not change the wettability of the PMMA surface in this work.

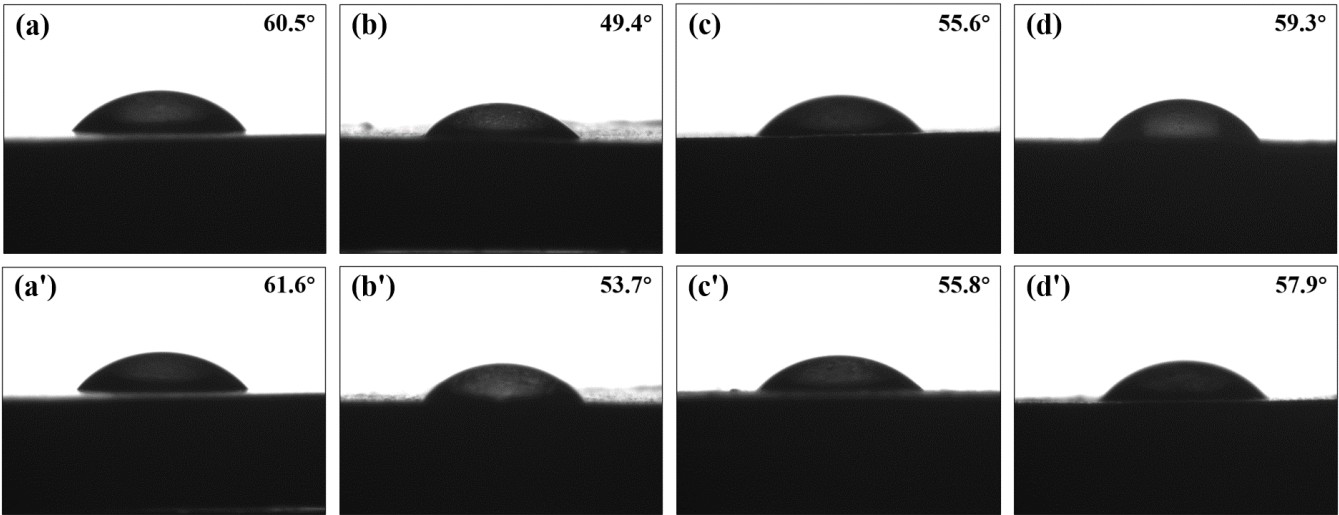

**Figure 5.** Contact angles of (**a**,**a'**) PMMA substrate, DLC films deposited on Ti/PMMA with conductive material at the different bias of (**b**,**b'**) 550 V, (**c**,**c'**) 750 V, and (**d**,**d'**) 950 V. (**a**–**d**) represents the contact angle of the corresponding sample center, while (**a'**–**d'**) represents the contact angle of the corresponding sample edge.

### 3.3. Mechanical Hardness and Adhesion

Figure 6 demonstrates the mechanical hardness of the film deposited on Ti/PMMA at the different bias with and without conductive material. For the DLC950V-BD film, the hardness of A1 in the center of the sample is the smallest ($2.15 \pm 0.15$ GPa). Then, the hardness of the films increased significantly to 7.9 GPa with the test position getting closer to the edge. Raman analysis shows that the film at the center of the sample is a polymer-like film, so the hardness is small. While the film on the edge of the sample is the DLC film, so the hardness is higher.

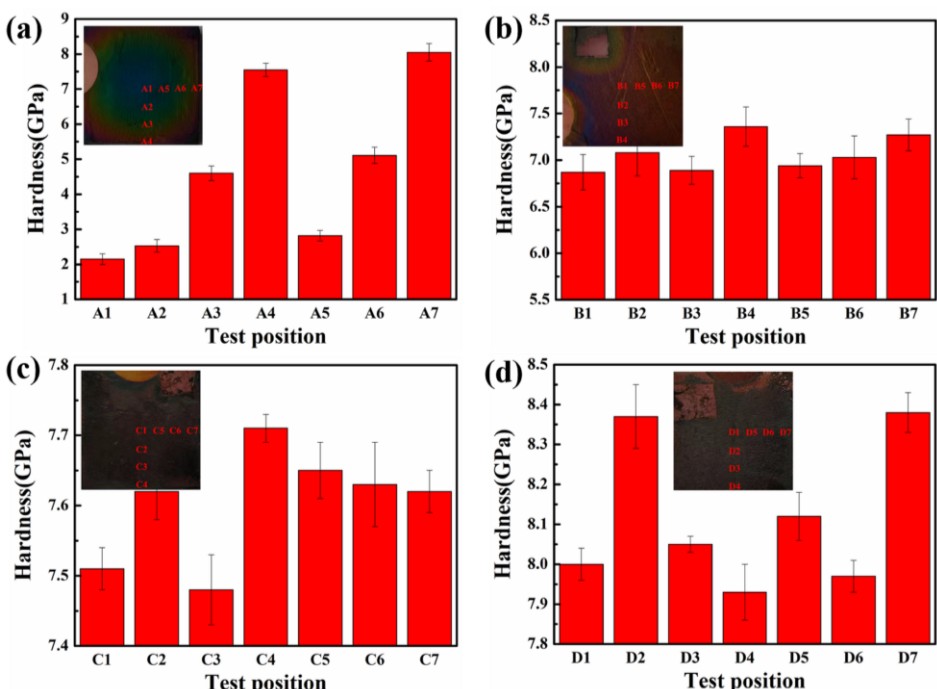

**Figure 6.** Mechanical hardness of (**a**) the film deposited on Ti/PMMA without conductive material at the bias of 950 V, and DLC films deposited on Ti/PMMA at a different substrate bias of (**b**) 550 V, (**c**) 750 V, and (**d**) 950 V with conductive material.

For the DLC550V-D film, the hardness of B1 in the center of the sample is $6.87 \pm 0.19$ GPa, the hardness of B2 and B5 points tested slightly towards the edge is $7.08 \pm 0.25$ and $6.94 \pm 0.13$, respectively; Subsequently, the hardness of the film almost did not change ($7.3 \pm 0.2$ GPa) as the test continued towards the edge of the sample. It can be seen that the hardness of different positions of the film is relatively uniform, namely, the hardness of the DLC550V-D film is $7.1 \pm 0.2$ GPa. With the increase of bias voltage, the hardness of the film center and edge is basically constant ($7.6 \pm 0.1$ GPa for the DLC750V-D film and $8.2 \pm 0.2$ GPa for the DLC950V-D film, respectively). The hardness of the center and edge of the sample is basically constant because of the uniform structure of the film. However, it can be seen that the hardness of the films increases slightly with the increase of substrate bias, which may be mainly attributed to the ion bombardment. With the increase of bias voltage, the energy of ion bombardment on the film surface increases. Higher energetic ion bombardment was expected to etch the weak bond and increase the film density, and while denser films carry much higher load, it is reasonable that the hardness increasing with the increase of bias voltages is closely related to the increase of films density.

The adhesion strength of films on the PMMA substrate was evaluated by the scratch test. In the scratch test, as the diamond tip moved slowly against the film, the removal of film would progressively occur. The load at which the film was peeled off completely from the substrate was recorded as the critical load, and a sharp increase in friction force was observed thereat. Figure 7 shows the scratch curves of DLC films deposited on Ti/PMMA at a different substrate bias with conductive material, where friction force was recorded as a function of loading force. As shown in Figure 7, it can be seen that the friction force of all films has gone through three distinct stages with the increase of loading force: steady increase stage (0 to Lc1), sudden change stage (Lc1 to Lc2), and failure stage ($\geq$Lc2). The first stage is the peeling-off of the DLC top film, the second stage is the peeling-off of the Ti interlayer, and the third stage is the failure stage, which means that the diamond tip has scratched the PMMA substrate.

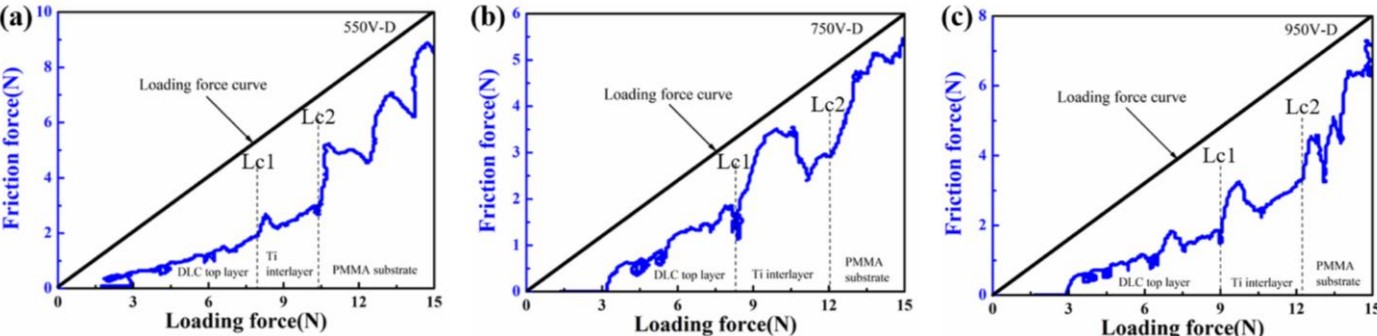

**Figure 7.** The scratch curves of DLC films deposited on Ti/PMMA at different substrate bias of (**a**) 550 V, (**b**) 750 V, and (**c**) 950 V with conductive material.

The critical force for peeling the DLC film is 7.9 N, and the critical force for peeling the Ti interlayer is 10.4 N for the DLC550V-D film. With the increase of bias voltage, the critical force for peeling the DLC film slightly increases to 8.3 N, and the critical force for peeling the Ti layer slightly increases to 12 N for the DLC550V-D film. As the bias voltage further increases to 950 V, the critical force for peeling the DLC film increases to 9 N, and the critical force for peeling off the Ti layer increases to 12.2 N for the DLC950V-D film. It is obvious that the critical force for peeling the DLC mainly depends on the thickness and hardness of the film, and its change is mainly positively related to hardness when the thickness is the same. Theoretically, the critical force for peeling the Ti interlayer should be the same due to the same deposition parameters. The difference in here may be mainly attributed to the measurement error due to the existence of a large number of micro-bumps on the PMMA surface. Based on the above analysis, the adhesion of the DLC film on PMMA ultimately depends on the hardness of the DLC top layer in fact. However, it is worth mentioning that the addition of the Ti interlayer between DLC and PMMA means the addition of another adhesive interface. That is, the adhesion interface has transferred from the DLC/PMMA interface to the DLC/Ti interface and Ti/PMMA interface, which virtually increases the risk of peeling off of the film. In this work, the Ti layer was introduced as a conductive layer in order to obtain DLC films on PMMA with uniform structure and performance, and the adhesion problem has not received enough attention. In the following research, the adhesion of films will be studied in detail on the basis of preparing uniform films on PMMA.

### 3.4. Tribological Properties

Figure 8 presents the friction coefficient curves of PMMA, Ti/PMMA, and the films prepared on Ti/PMMA at a different substrate bias without conductive materials. As can be seen, for the PMMA substrate, the friction coefficient increases quickly and reaches the stable values of ~0.5 after about 150 laps. Furthermore, the friction coefficient of all other samples tends to be the same and is equal to the friction coefficient of the virgin PMMA after the different laps tribotests. The morphology of the corresponding wear track is shown in Figure 9. For the PMMA substrate, a lot of micro bumps derived from the manufacturing process could be observed on its surface, and these micro bumps are all polished after 30 min tribotests. For all other samples, the wear track morphology is basically the same as that of the original PMMA, which indicates that all films deposited without conductive materials are worn off, which is mainly attributed to the polymer-like structural characteristics of the films confirmed by Raman analysis.

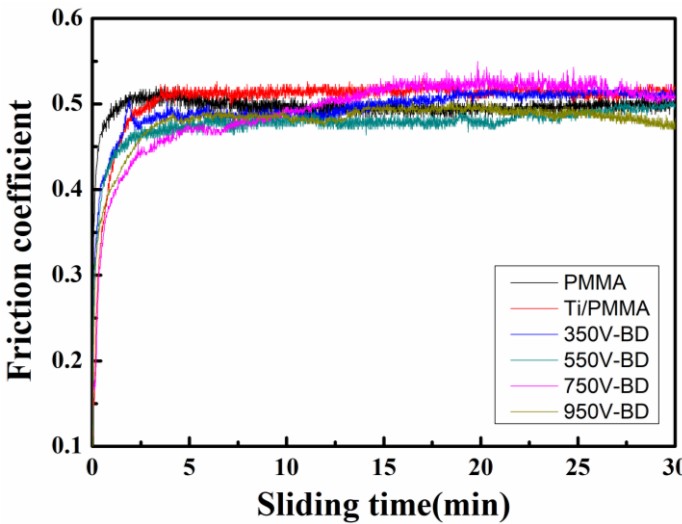

**Figure 8.** Friction coefficient curves of PMMA, Ti/PMMA, and the films prepared on Ti/PMMA at a different substrate bias without conductive material.

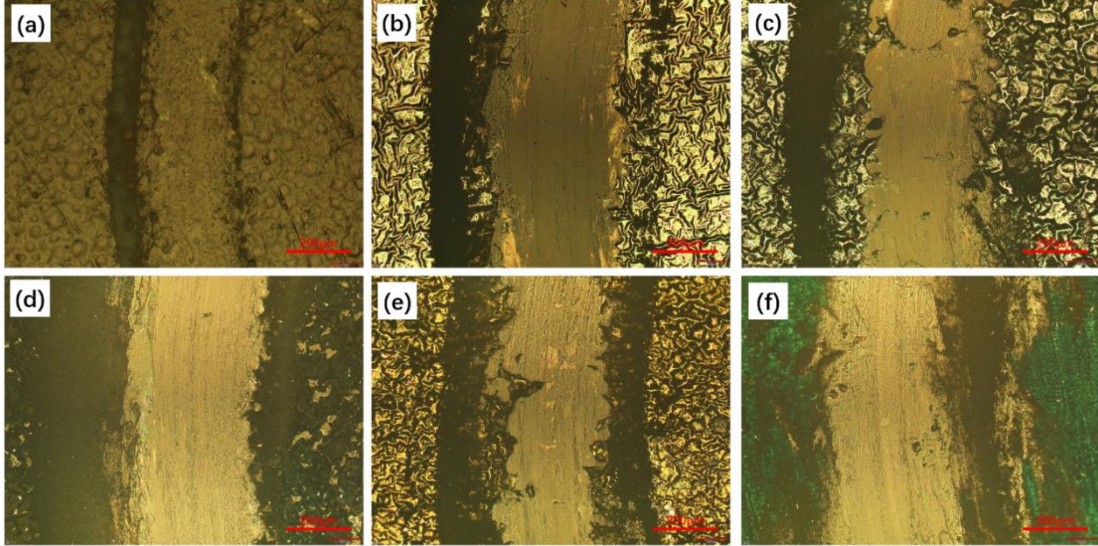

**Figure 9.** Optical images of wear tracks of (**a**) PMMA, (**b**) Ti/PMMA, and the films prepared on Ti/PMMA at different substrate bias of (**c**) 350 V, (**d**) 550 V, (**e**) 750 V, and (**f**) 950 V without conductive material.

However, a completely different result could be showed for the films deposited with conductive material. Figure 10 displays the friction coefficient curves of PMMA and the films prepared on Ti/PMMA at a different substrate bias with conductive materials. It can be seen that the PMMA substrate exhibits a relatively high friction coefficient of ~0.50. As the bias increases, the friction coefficient of the DLC350V-D film is the same as that of the original PMMA after 1500 cycles tribotests, which is mainly due to the deposition failure of DLC films on Ti/PMMA, while the Ti interlayer is worn through after 1500 laps. As the bias increases to 550 V, the DLC550V-D film presents a lower friction coefficient of 0.46 than that of the original PMMA and its friction coefficient increases monotonously from the initial friction coefficient of about 0.25 to 0.5 after 4500 laps tribotests and remained stable value of ~0.5. As the bias increases further from 550 V to 750 V, the DLC750V-D film displays an obviously lower friction coefficient (~0.36) than that of the DLC550V-D film. As the bias increases finally to 950 V, the DLC950V-D film displays the lowest friction coefficient (~0.32).

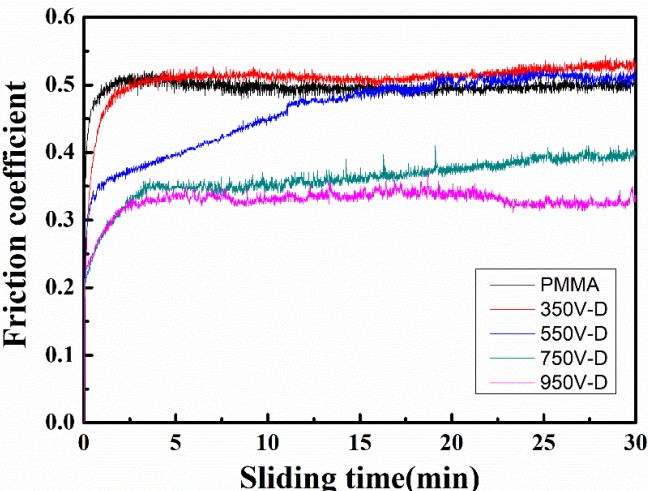

**Figure 10.** Friction coefficient curves of PMMA and the films prepared on Ti/PMMA at a different substrate bias with conductive materials.

Figure 11 presents the optical images of wear tracks of the DLC films prepared on Ti/PMMA at a different substrate bias with conductive materials. For the DLC350V-D film, the wear track morphology is basically the same as that of the original PMMA, and a large amount of wear debris accumulates on both sides of the wear tracks, which indicates the worn off of the film. For the DLC550V-D film, the film was worn through after 4500 cycles, confirmed by the wear track and friction coefficient curve. As the bias increases further to 750 V, the width of the wear track becomes significantly narrower, and the micro bumps on the sample surface are not completely polished, and the "gullies" on the wear track can still be clearly observed. This shows that the wear resistance of the sample is obviously enhanced. As the bias increases finally to 950 V, the wear track morphology of the DLC950V-D film is obviously different from that of PMMA. The wear only occurs on the top of the micro bumps on the surface of the sample, which indicates that the wear resistance of the DLC950V-D film is optimal. This is mainly attributed to the higher hardness of the DLC950V-D film than that of all other films. The above experimental results show that the wear resistance of PMMA can be improved significantly by preparing DLC films on it.

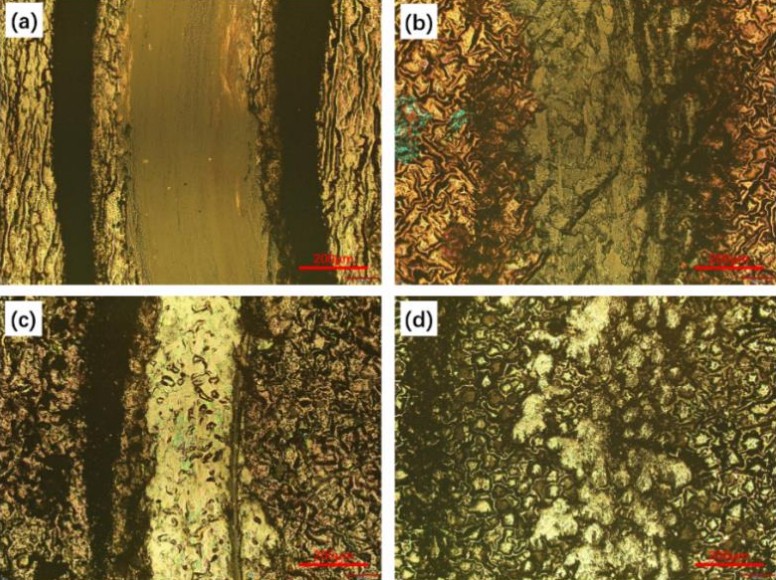

**Figure 11.** Optical images of the wear tracks of the DLC films prepared on Ti/PMMA at a different substrate bias of (**a**) 350 V, (**b**) 550 V, (**c**) 750 V, and (**d**) 950 V with conductive materials.

## 4. Conclusions

Previous experiments show that it is difficult for DLC film to be directly deposited on PMMA due to its good insulation. Therefore, the conductive Ti layer was introduced as an interlayer in this work, and the influence of substrate bias on the growth mechanism, microstructure, and tribological behavior of films on Ti/PMMA with and without conductive material was investigated systematically. The results indicate that the DLC characteristics of the films without conductive material could only be measured at the edge of the sample, even when the substrate bias is up to 950 V during deposition. For the films deposited with conductive materials, the films exhibit uniform DLC structure and mechanical hardness when the bias voltage is only 550 V. These results mainly depend on the difference of deposition mechanism of films on Ti/PMMA with or without conductive materials. Furthermore, the deposited DLC does not change the wettability of PMMA, while the addition of the Ti interlayer virtually increases the risk of peeling off of films because of the addition of extra adhesive interface. The results of the tribological study show that the films on Ti/PMMA without conductive material show poor wear resistance, while the films deposited with conductive material have a higher wear resistance, which is mainly attributed to their higher hardness. This research work can provide basic theoretical guidance for depositing uniform DLC films on PMMA and even all non-conductive substrates by the PECVD method.

**Author Contributions:** Conceptualization, Y.B. and L.Q.; methodology, Y.B. and K.Y.; software, B.Z. and L.Q.; validation, B.Z. and L.Q.; formal analysis, B.Z. and L.Q.; investigation, Y.B. and L.Q.; resources, Y.B.; data curation, Y.B. and L.Q.; writing—original draft preparation, Y.B.; writing—review and editing, B.K. and L.Q.; supervision, Y.B. and B.K.; project administration, Y.B. and B.K.; funding acquisition, Y.B. All authors have read and agreed to the published version of the manuscript.

**Funding:** This research was funded by the Scientific Research Project of Lanzhou science and technology bureau (2021-1-114).

**Institutional Review Board Statement:** Not applicable.

**Informed Consent Statement:** Not applicable.

**Data Availability Statement:** The data are available upon reasonable request from the corresponding author.

**Acknowledgments:** The authors are grateful to the Scientific Research Project of Lanzhou science and technology bureau (2021-1-114) for the financial support. They also wish to thank especially Chaojie Sun for testing the samples. The help rendered by the other colleagues and students in writing assistance and other aspects is also gratefully acknowledged.

**Conflicts of Interest:** The authors declare no conflict of interest.

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
