# Peer review of "A Simple and Effective Method to Adjust the Structure and Performance of DLC Films on Polymethyl Methacrylate(PMMA) Substrate"

_coatings, doi:10.3390/coatings13020320_

Round 1

Reviewer 1 Report

I have reviewed the manuscript coatings-2155272-peer-review-v1. The aim of the work is clear, and authors employed proper method. set-up and characterization. The work is interesting for from academic and technological aspects.  The manuscript is well written, however, for the sake of improvement. I suggest authors responded to the hereafter comments. Hence, the manuscript is acceptable for publication.

1)      Page 2, line 88, what do you mean with (conducting material): You have PMMA, Ti/PMMA and you working to deposit DLC, so what do you meant with “Conducting material. Better to include schematic diagram for the intended Film composite in

2)      In Fig.2, it would better to make base line correction and smooth Raman signals. And enlarge the wavenumber range 1000 to 2000 cm-1

3)      Added labels (a), (b), etc.. the charts in Fig.2

4)      This Sentence “For the Ti layer, no Raman peak can be observed, because Raman spectrum is molecular spectrum based on molecular vibration, while metal is atomic structure (no molecular vibration) “ is not correct. Raman Bands of Ti can be observed, around below 600 cm -1, but their intensity is week.

5)      Identify the D and G peaks characteristics to DLC film in Fig. 2

6)      Do smoothing to the signal of Fig.3 and identify the D and G peaks in the chart.

7)      Your samples area is small 20 x 20 mm2, and you working with low pressure plasma, the plasma is uniform for such small area, hence any discussion about uniformity is meaningless

Author Response

Thank you so much for your comments. We have corrected our manuscript according to your suggestions already. Moreover, the resolution of all pictures has been improved and the previous pictures have been replaced in this revision. The changes are listed here.

According to the editor's and reviewer's comments:

Reviewer 1

I have reviewed the manuscript coatings-2155272-peer-review-v1. The aim of the work is clear, and authors employed proper method. set-up and characterization. The work is interesting for from academic and technological aspects. The manuscript is well written, however, for the sake of improvement. I suggest authors responded to the hereafter comments. Hence, the manuscript is acceptable for publication.

1) Page 2, line 88, what do you mean with (conducting material): You have PMMA, Ti/PMMA and you working to deposit DLC, so what do you meant with “Conducting material. Better to include schematic diagram for the intended Film composite in.

Correction: Thank you for your suggestion. In the previous experiment, we found that it was difficult to directly deposit DLC film on the surface of PMMA. We speculated that PMMA is an insulating material, so it is difficult to form an electric potential difference between the top electrode and PMMA. In order to verify it, Ti metal layer was introduced as an interlayer. However, it is still difficult to form an electric potential difference between the top electrode and Ti/PMMA, because PMMA prevented the direct connection between Ti interlayer and the sample holder. Therefore, we set an electric conductor (called “conductive material” in this work, as shown in the dotted red line area below) between the Ti metal layer and the sample holder to connect with each other.

2) In Fig.2, it would better to make base line correction and smooth Raman signals. And enlarge the wavenumber range 1000 to 2000 cm-1.

Correction: Thank you for your suggestion. The Fig.2 has been improved and added in the text.

3) Added labels (a), (b), etc.. the charts in Fig.2.

Correction: Thank you for your suggestion. We didn't express clearly the meaning of Fig. 2. In Fig. 2, the color of the Raman curve represents the results measured at the corresponding color position in the illustration. For example, the red Raman curve represents the data measured at the red position in the illustration.

4) This Sentence “For the Ti layer, no Raman peak can be observed, because Raman spectrum is molecular spectrum based on molecular vibration, while metal is atomic structure (no molecular vibration) “is not correct. Raman Bands of Ti can be observed, around below 600 cm-1, but their intensity is week.

Correction: Thank you for your suggestion. The related sentence has been corrected.

5) Identify the D and G peaks characteristics to DLC film in Fig. 2.

Correction: Thank you for your suggestion. The D and G peak of DLC film has been fitted as shown in Fig.2f.

6) Do smoothing to the signal of Fig.3 and identify the D and G peaks in the chart.

Correction: Thank you for your suggestion. The Fig.3 has been improved and added in the text.

7) Your samples area is small 20 x 20 mm2, and you working with low pressure plasma, the plasma is uniform for such small area, hence any discussion about uniformity is meaningless.

Correction: Thank you for your suggestion. At first, we also thought that the plasma distribution on the surface of such a small sample should not be too different. However, we did find the nonuniformity of the structure and performance of the DLC on PMMA during performance test, so we speculated that it was caused by the nonuniformity of plasma. Therefore, the experimental scheme of this work was designed to confirm it.

The authors are very grateful to the editors and reviewers for their comments or suggestions, which are very useful for improving the quality of our paper. We are looking forward to your response, and any kind of information or suggestion will be greatly appreciated.

Sincerely yours,

Yinzhong Bu

Reviewer 2 Report

1- Line 13: Please write DLC in the open form first before the abbreviation.

Lines 13-18: Please make this part of the summary more understandable. There seems to be a repetition here.

2- Page 1, line 29: Please check the tense of "...PMMA has used widely..."

3- Page 2, line 56: Does DLC film improve the surface properties or mechanical properties of PMMA?

4-Page 2- line 66: What is PECVD method?

5- Page 2 lines 88-89: This sentence is not clear. Is it meant Ti as the conductive material? What was it meant by some samples?

6- Fig. 1 should be explained in more detail. Where is the conductive material, where is the steel holder, and what type of conductive material was used, should be explained; it is not clear.

7- Page 3, line 108: Is it mechanical properties or only hardness measured by nanoindenter?

8- Fig. 2: The numbering is missing.

9- Page 7, lines 219-220: "As a result, DLC films with the uniform structure were successfully prepared on PMMA under a higher bias (≥550V)." It should be explained that the DLC is formed not directly over PMMA, but over the interlayer of Ti.

10- Page 7, lines 219-220: "As a result, DLC films with the uniform structure were successfully prepared on PMMA under a higher bias (≥550V)." It should be explained that the DLC is formed not directly over PMMA, but over the interlayer of Ti.

11- Page 9, lines 290-292: Normally, the peeling force should depend on the adherence between the layers and it seems that the higher the hardness of DLC layer higher the adherence.

12- Page 9, Lines 301-304: How was it decided that the peeling adherence of DLC on the substrate is not enough?

Author Response

Thank you so much for your comments. We have corrected our manuscript according to your suggestions already. Moreover, the resolution of all pictures has been improved and the previous pictures have been replaced in this revision. The changes are listed here.

According to the editor's and reviewer's comments:

Reviewer 2

1- Line 13: Please write DLC in the open form first before the abbreviation. Lines 13-18: Please make this part of the summary more understandable. There seems to be a repetition here.

Correction: Thank you for your suggestion. The related content has been improved.

2- Page 1, line 29: Please check the tense of "...PMMA has used widely..."

Correction: Thank you for your suggestion. The related express has been improved.

3- Page 2, line 56: Does DLC film improve the surface properties or mechanical properties of PMMA?

Correction: Thank you for your suggestion. Yes, the hardness of DLC film reaches 7-8GPa, which is far greater than the mechanical hardness of PMMA surface (25.9/HV0.2, P<0.001).

4-Page 2- line 66: What is PECVD method?

Correction: Thank you for your suggestion. All films were deposited by plasma-enhanced chemical vapor deposition (PECVD) technology. The related sentence has been improved and added in the text.

5- Page 2 lines 88-89: This sentence is not clear. Is it meant Ti as the conductive material? What was it meant by some samples?

Correction: Thank you for your suggestion. In the previous experiment, we found that it was difficult to directly deposit DLC film on the surface of PMMA. We speculated that PMMA is an insulating material, so it is difficult to form an electric potential difference between the top electrode and PMMA. In order to verify it, Ti metal layer was introduced as an interlayer. However, it is still difficult to form an electric potential difference between the top electrode and Ti/PMMA, because PMMA prevented the direct connection between Ti interlayer and the sample holder. Therefore, we set an electric conductor (called “conductive material” in this work, as shown in the dotted red line area below) between the Ti metal layer and the sample holder to connect with each other.

At each deposition, twenty samples were placed on the sample holder at the same time in this work, and ten of them used conductive materials and others not.

6- Fig. 1 should be explained in more detail. Where is the conductive material, where is the steel holder, and what type of conductive material was used, should be explained; it is not clear.

Correction: Thank you for your suggestion. Silver conductive adhesives was used a conductive material in this work (as shown in the dotted red line area below), steel holder was shown in the dotted green line area below.

7- Page 3, line 108: Is it mechanical properties or only hardness measured by nanoindenter?

Correction: Thank you for your suggestion. Only hardness of films was measured in this work.

8- Fig. 2: The numbering is missing.

Correction: Thank you for your suggestion. We didn't express clearly the meaning of Fig. 2. In Figure 2, the color of the Raman curve represents the results measured at the corresponding color position in the illustration. For example, the red Raman curve represents the data measured at the red position in the illustration.

9- Page 7, lines 219-220: "As a result, DLC films with the uniform structure were successfully prepared on PMMA under a higher bias (≥550V)." It should be explained that the DLC is formed not directly over PMMA, but over the interlayer of Ti.

Correction: Thank you for your suggestion. When the bias voltage is ≤350V, the electric field between the two plates is not enough to generate glow discharge to ionize the working gas, this is confirmed by the fact that glow is hardly observed through the viewpoint on the vacuum chamber. Namely, the gas is eliminated without ionization, leading to the failure of film deposition. This is the main reason why it is impossible to deposit films on Ti/PMMA with or without conductive material under a lower bias. For the films deposited with conductive material, because the conductive material connects the Ti layer with the steel sample holder (as shown in Fig.4b), the electric potential field strength between top plate and Ti/PMMA is no longer constant, but increases with the increase of bias voltage due to the good conductivity of Ti interlayer. That is, the ion energy between top plate and Ti/PMMA is almost the same as that between top plate and steel sample holder, and there is no difference in the ion energy reaching the center and edge of the sample.

10- Page 7, lines 219-220: "As a result, DLC films with the uniform structure were successfully prepared on PMMA under a higher bias (≥550V)." It should be explained that the DLC is formed not directly over PMMA, but over the interlayer of Ti.

Correction: Thank you for your suggestion. When the bias voltage is ≤350V, the electric field between the two plates is not enough to generate glow discharge to ionize the working gas, this is confirmed by the fact that glow is hardly observed through the viewpoint on the vacuum chamber. Namely, the gas is eliminated without ionization, leading to the failure of film deposition. This is the main reason why it is impossible to deposit films on Ti/PMMA with or without conductive material under a lower bias. For the films deposited with conductive material, because the conductive material connects the Ti layer with the steel sample holder (as shown in Fig.4b), the electric potential field strength between top plate and Ti/PMMA is no longer constant, but increases with the increase of bias voltage due to the good conductivity of Ti interlayer. That is, the ion energy between top plate and Ti/PMMA is almost the same as that between top plate and steel sample holder, and there is no difference in the ion energy reaching the center and edge of the sample.

11- Page 9, lines 290-292: Normally, the peeling force should depend on the adherence between the layers and it seems that the higher the hardness of DLC layer higher the adherence.

Correction: Thank you for your suggestion. In the scratch test, higher hardness may mean higher bearing capacity for the diamond tip, which implies that the diamond indenter needs more lateral force to peels off the harder film.

12- Page 9, Lines 301-304: How was it decided that the peeling adherence of DLC on the substrate is not enough?

Correction: Thank you for your suggestion. The author doesn't quite understand the meaning of this comment. In this work, Ti layer was introduced as a conductive layer in order to obtain DLC films on PMMA with uniform structure and performance, but it does not play a positive role in improving the adhesion of the film on PMMA. Moreover, we do not focus on how to improve the adhesion of films on PMMA, it will be studied in more detail on the basis of preparing uniform films on PMMA in the following research.

The authors are very grateful to the editors and reviewers for their comments or suggestions, which are very useful for improving the quality of our paper. We are looking forward to your response, and any kind of information or suggestion will be greatly appreciated.

Sincerely yours,

Yinzhong Bu

Reviewer 3 Report

--The authors should include the advantage of the proposed work

-- How the principal purpose of these differences of films on Ti/PMMA with and without conductive material is better compared to the other conductive material?

--The author should incorporate mathematical equations. 

Author Response

Thank you so much for your comments. We have corrected our manuscript according to your suggestions already. Moreover, the resolution of all pictures has been improved and the previous pictures have been replaced in this revision. The changes are listed here.

According to the editor's and reviewer's comments:

Reviewer 3

  1. The authors should include the advantage of the proposed work.

Correction: Thank you for your suggestion. This research work can provide a simple and effective method and a basic theoretical guidance for depositing DLC films with uniform structure and performance on PMMA and even all non-conductive substrate. This may be the advantage of this work.

  1. How the principal purpose of these differences of films on Ti/PMMA with and without conductive material is better compared to the other conductive material?

Correction: Thank you for your suggestion. The author doesn't quite understand the meaning of this comment. No matter what kind of conductive material, theoretically, it can achieve the purpose of this work as long as this material is conductive. In other words, the type of conductive materials will not cause differences in the structure and performance of the film in theory.

  1. The author should incorporate mathematical equations.

Correction: Thank you for your suggestion. To be honest, the authors does not sure how to combine the deposition mechanism with the mathematical equations, and the authors are not really good at mathematics. Indeed, the learning of professional knowledge of mathematics should be strengthened.

The authors are very grateful to the editors and reviewers for their comments or suggestions, which are very useful for improving the quality of our paper. We are looking forward to your response, and any kind of information or suggestion will be greatly appreciated.

Sincerely yours,

Yinzhong Bu

Round 2

Reviewer 2 Report

The necessary improvements were made. It can be accepted.

Reviewer 3 Report

Nil